# Bowel Health in U.S. Shift Workers: Insights from a Cross-Sectional Study (NHANES)

**DOI:** 10.3390/ijerph19063334

**Published:** 2022-03-11

**Authors:** Maximilian Andreas Storz, Mauro Lombardo, Gianluca Rizzo, Alexander Müller, Ann-Kathrin Lederer

**Affiliations:** 1Center for Complementary Medicine, Department of Internal Medicine II, Freiburg University Hospital, Faculty of Medicine, University of Freiburg, 79106 Freiburg, Germany; alexander.mueller@uniklinik-freiburg.de (A.M.); ann-kathrin.lederer@uniklinik-freiburg.de (A.-K.L.); 2Department of Human Sciences and Promotion of the Quality of Life, San Raffaele Roma Open University, 00166 Rome, Italy; mauro.lombardo@uniroma5.it; 3Independent Researcher, Via Venezuela 66, 98121 Messina, Italy; gianlucarizzo@email.it; 4Department of General, Visceral and Transplantation Surgery, University Medical Center of the Johannes Gutenberg-University Mainz, 55116 Mainz, Germany

**Keywords:** diet, nutrition, shiftwork, work schedule, NHANES, bowel health, gastrointestinal symptoms, constipation, diarrhea

## Abstract

Working outside of regular daytime hours is increasingly common in current societies and poses a substantial challenge to an individual’s biological rhythm. Disruptions of the gastrointestinal tract’s circadian rhythm and poor dietary choices subsequent to shiftwork may predispose the shift workforce to an increased risk of gastrointestinal disorders, including constipation, peptic ulcer disease, and erosive gastritis. We investigated bowel health in a US population of shift workers, using data from the National Health and Nutrition Examination Survey, and compared bowel movement (BM) frequency and defecation patterns between 2007 day workers and 458 shift workers (representing 55,305,037 US workers). Using bivariate and multivariate logistic regression techniques, our results suggested no association between shiftwork status and BM frequency, bowel leakage of gas, and stool consistency. Constipation prevalence was high but comparable in both groups (6.90% vs. 7.09%). The low fiber intake observed in both groups (15.07 vs. 16.75 g/day) could play a potential role here. The two groups did not differ with regard to other nutrients that may influence BM frequency and stool consistency (e.g., carbohydrate or caffeine intake). Additional studies including food group analyses and fecal biomarkers are warranted for a better understanding of GI health in shift workers.

## 1. Introduction

Working outside of regular daytime hours is increasingly common in modern societies [1]. The increasing need for services operating on a 24 h basis, particularly in healthcare and transport, causes workers to routinely work on the basis of shift schedules [2]. Shiftwork and rotating shift schedules are a challenge to an individual’s biological rhythms and lead to alterations in the biological clock [2,3].

Subsequent disruptions in the body’s circadian rhythm may negatively affect psychosocial wellbeing and nutrition intake [3]. Due to frequently changing day routines, shift workers are often forced to eat on an irregular basis [4,5]. The result is an increased consumption of meals that are high in fat, sugar, processed grains, and animal products (dairy and meat in particular) [3,6,7].

It is now widely accepted that the gastrointestinal (GI) tract has a circadian rhythm that influences bowel movements, secretion of gastric juices, and appetite regulation [8,9,10]. Disruptions of the GI tract’s circadian rhythm, in combination with the aforementioned poor dietary choices in shift workers, may predispose this population to an increased risk of gastrointestinal disorders [3,11].

Knutson and Boggild reviewed GI disorders among shift workers and found an increased risk of GI symptoms and peptic ulcer disease in this population [8]. Several other studies also demonstrated positive associations between shiftwork and GI disorders, as summarized in Figure 1 (based on [12,13,14,15,16]).

A more recent systematic review and meta-analysis by Chang and Peng essentially confirmed the results of Knutson and Boggild [3]. This analysis included 16 studies, mostly from Asia (Japan and Korea) and the Middle East (Iran, Kuwait, Saudi Arabia, and Egypt). Studies investigating GI health in North American shift-working populations, however, are scarce, and limited to a small handful of trials [17].

The present study sought to address this lack of studies and investigated bowel health in a United States-based population of shift workers. Using data from the National Health and Nutrition Examination Survey (NHANES), we compared bowel health in self-reported shift workers and day workers, focusing on bowel movement (BM) frequency and defecation patterns.

## 2. Materials and Methods

### 2.1. NHANES Characteristics

This is a cross-sectional study with publicly available data from the National Health and Nutrition Examination Survey (NHANES) [18]. The NHANES is a large U.S.-based, multistage, stratified survey conducted by the Centers for Disease Control and Prevention (CDC) [18,19]. NHANES was designed to systematically collect health and nutrition data on the U.S. population, and to monitor the health of U.S. citizens [20].

This ongoing program is funded by the National Center for Health Statistics (NCHS) of the CDC [21]. NHANES is one of the world’s largest cross-sectional studies in terms of study size, diversity, and data accessibility [22], and data from the NHANES has often been used to analyze health-related questions in shift-working populations [23,24,25,26,27,28]. The NHANES is a nationally representative survey for the noninstitutionalized U.S. civilian population of all ages residing in all 50 states and Washington D.C. [29]. A key feature of the NHANES is the complex, multistage, probability sampling design that includes multiple study locations. Background information on the sampling design itself, and on oversampling of certain ethnic groups, is publicly available and may be obtained from the National Health and Nutrition Examination Survey Sample Design Handbook (2007–2010) [30].

For this study, we combined two NHANES cycles (the 2007–2008 cycle and the 2009–2010 cycle) [31,32]. Before the appending procedure, we ensured that the used variables did not differ from cycle to cycle (in terms of both wording and categorization). More recent NHANES cycles (e.g., 2010–2011 and onward) were not appended because some key data on occupational health were no longer included in these circles. More than 10,000 individuals participated in the NHANES per cycle [33]. Informed consent was obtained from all participants, and all procedures in the NHANES were in accordance with the Declaration of Helsinki [34,35].

### 2.2. Study Population, Outcome, and Exposure

Demographic, examination, and interview data were used for this particular analysis. The different datasets employed for this investigation are explained hereafter in detail. All modules were merged to produce a single dataset that contained all of the relevant information.

#### 2.2.1. Occupational Health

We assessed shiftwork status using data from the Occupation Questionnaire Section, which covers employment and other important variables relating to the daily work environment [36,37]. The question entitled “Which of the following best describes the hours you usually work at your main job or business?” was used to assess shiftwork status. Potential answers included (1) a regular daytime schedule, (2) a regular evening shift, (3) a regular night shift, (4) a rotating shift, and (5) another schedule. We removed the last answer (5) because this particular schedule was not described in greater detail [38]. For our analysis, evening/night shift and rotating shift were combined into one group (based on a study by Wirth et al. [38]). This group is hereafter named shift workers. Shift workers were compared to individuals that chose answer (1) (a regular daytime schedule), when asked the aforementioned question. This group is hereafter named day workers. We limited the analysis to two NHANES cycles (2007/2008 and 2009/2010), because the aforementioned work schedule question was removed in subsequent NHANES cycles, making appending and comparison of data difficult [39].

#### 2.2.2. Demographic Data

We obtained participants’ demographic data from the demographics public release file [40,41]. Demographic data covered sex (female and male), age, race/ethnicity, education level, marital status, and annual household income. Predefined NHANES variables and categories were not changed, with two exceptions (marital status and household income). Race/ethnicity comprised five subcategories, including “Mexican American”, “non-Hispanic White”, “non-Hispanic Black”, “other Hispanic”, and “other race” (includes mixed race). Marital status comprised three categories: married or living with a partner, widowed/divorced/separated, and never married. Annual household income included two categories: over USD 20,000 and under USD 20,000.

#### 2.2.3. Dietary and Examination Data

Dietary data included daily calorie intake and macronutrient intake (fat, carbohydrate, and protein) [42,43]. Moreover, we specifically investigated saturated fat intake, monounsaturated fat intake, and polyunsaturated fat intake. Additional data also included fiber intake, alcohol intake, and coffee intake. Nutrients were not adjusted for total calorie intake because a preliminary analysis revealed no significant intergroup differences in daily total caloric intake. We also investigated daily aggregates of water (moisture), which included all moisture present in beverages and foods, including tap water and bottled water.

Examination data included body mass index (BMI) [44,45], which we categorized into four groups, including obesity (BMI ≥ 30 kg/m^2^), overweight (BMI 25–29.99 kg/m^2^), normal weight (BMI 18.5–24.99 kg/m^2^), and underweight (BMI ≤ 18.49 kg/m^2^). We categorized smoking status using the approach of O’Neil et al. [46].

#### 2.2.4. Bowel Health

We assessed bowel health using data from the Bowel Health Questionnaire (BHQ) [47,48]. A detailed description of the methods can be obtained from one of our previous publications [49]. The BHQ provides personal interview data on stool morphology, defecating function, and incontinence symptoms in adults aged 20 years and older. The 2007/2008 NHANES bowel component included six questions. Four questions covered adult incontinence leakage, based on Rockwood’s Fecal Incontinence Severity Index (FISI) [50]. The four symptoms composing the index are incontinence of mucus, liquid stool, solid stool, and gas [47,48]. Stool consistency was assessed with the Bristol Stool Form Scale (BSFS) [51], a scale that has been used in a series of clinical trials to assess stool form in gastrointestinal disorders [52]. In the field of gastroenterology, the BSFS is considered a reliable and valid tool to assess stool form [53].

Participants were asked to define their stool by recording the number type (BSFS type 1–7) that corresponded to their usual/most common stool type. The BSFS demonstrated a modest correlation with colonic transit time (CTT) in a previous study, and a BSFS of ≤2 has been suggested as a surrogate marker for a delayed CTT in Westerners [54]. As a matter of fact, studies also suggested that the BSFS score correlates with the severity of incontinence [55]. As such, we defined constipation and diarrhea based on self-reported typical stool type. Following the classification method of Wilson [56], stool types 6 (fluffy pieces with ragged edges, a mushy stool) and 7 (watery, no solid pieces) were defined as diarrhea. In contrast, stool types 1 (separate hard lumps, like nuts) and 2 (sausage-like, but lumpy) were defined as constipation. Moreover, we used Ditah’s approach to investigate symptoms of (fecal) incontinence (FI) [57]. The latter was defined as any kind of accidental leakage or involuntary loss of stool (liquid or solid) or mucus during the past month. Of note, this definition did not include gas leakage, which was analyzed in a separate step.

### 2.3. Statistical Analysis

We used STATA version 14 (StataCorp., College Stadion, TX, USA) for the entire statistical analysis [58]. Sample weights (provided with the NHANES files) were used to account for the complex, multistage, probability sampling design. The weighting of NHANES data allows for extrapolation of study findings to the U.S. national population [59] and takes into account the unequal probabilities of selection resulting from the sample design, non-response, and planned oversampling of certain population groups [60].

In accordance with the NHANES analytic guidelines, a 4-year weight (2007–2010) was generated to estimate reliable weighted percentages adjusted to the non-institutionalized US adult population [61]. All variables were compared between shift workers and day workers.

For the comparison of continuous and normally-distributed variables, we used the two-sample Student’s t-tests. We described these variables with their mean and the corresponding standard error in parenthesis. We compared categorical variables using STATA’s design-adjusted Rao–Scott test. This test is a design-adjusted version of the Pearson chi-square test. All categorical variables are reported in the following format: number of observations (weighted proportions (standard error)). In accordance with the current NCHS recommendations to report estimated proportions, we carefully assessed the reliability of all our estimated proportions [62]. To simplify that step, we used the STATA command “kg_nchs” [63]. This post-estimation command allows users to display a series of dichotomous flags that show whether the NCHS standards are met (or not). Proportions that do not meet the NCHS standard are flagged as “unreliable proportions” (for example, when the standard error exceeds 30% of the proportion estimate). We clearly marked these proportions in our tables using superscript letters. All statistical tests were two-sided, and statistical significance was determined at α = 0.05. 

We also ran multivariate regression models (standard binary, multinomial, and cumulative logistic regression models) to investigate potential associations between shiftwork status and stool patterns (as well as bowel movement frequency) after adjusting for age, sex, race/ethnicity, and income. Potential candidate predictors were chosen based on initial exploratory bivariate analyses. Apart from shiftwork status, only candidate predictors of interest and predictors with a bivariate relationship of significance *p* < 0.25 with the response variable were included in the multivariate logistic model.

In a first step, we used a multinomial logistic regression model to investigate potential associations between shiftwork and the BSFS-based stool patterns. In addition to shiftwork status, we considered sex, age, and income as important predictor variables. Age was treated as a categorical variable in this model. A normal stool pattern (defined as a BSFS > 2 and ≤ 5) was defined as the baseline category. Constipation was defined by a BSFS ≤ 2, whereas diarrhea was defined by a BSFS ≥ 6. Since we investigated clustered data, the number of clusters limited the maximum number of coefficients that could be simultaneously tested.

As such, we also made use of simple (binary) logistic regression models to investigate potential associations between shiftwork and stool patterns after adjusting for additional confounders (race/ethnicity). In a first model, we investigated whether shiftwork status was associated with higher odds for constipation (as defined by a BSFS ≤ 2) after adjusting for confounders. These covariates were also used in a third model, where we investigated whether shiftwork status was associated with higher odds for diarrhea (as defined by a BSFS ≥ 6).

Finally, we used a cumulative logistic regression model to investigate potential associations between shiftwork status and bowel movement frequency.

## 3. Results

A total of 13,435 participants completed the NHANES Occupation Questionnaire Section between 2007 and 2010 [36,37]. A total of 7002 participants were excluded (*n* = 3351 from the 2007/2008 cycle and *n* = 3651 from the 2009/2010 cycle) because of missing or inconclusive work hour descriptions (Figure 2). As described earlier in the methods section, we followed the approach of Wirth et al. and removed the category “another schedule” for a lack of more detailed description (*n* = 538) [38]. Our final sample included *n* = 2465 non-institutionalized participants after removal of all individuals with an incomplete dataset. This sample comprised *n* = 2007 day workers and *n* = 458 shift workers (Figure 2) and may be extrapolated to represent 55,305,037 U.S. workers.

Table 1 shows demographic and anthropometric characteristics of the study population stratified by shiftwork status. For categorical variables, Table 1 shows the number of observations and the weighted proportions (as well as the corresponding standard errors) in parentheses. For continuous and normally-distributed variables, data is shown as mean (and standard error in parentheses).

Our data suggested no significant differences between both groups with regard to sex. Shiftwork status, however, was not independent from race/ethnicity. The weighted proportion of non-Hispanic Blacks was significantly higher in shift workers (16.45% vs. 9.32%). In comparison, non-Hispanic Whites accounted for more than 70% of day workers. This proportion was significantly smaller in the shift worker group (63.77%).

In addition, our results revealed significant association between shiftwork status and marital status, education level, and annual household income, respectively. The weighted proportion of “never married” individuals was significantly higher in shift workers (33.55%) compared to day workers (15.81%). Moreover, the weighted proportion of married workers or workers living with a partner was significantly higher in the day worker group. We also observed a smaller (weighted) proportion of individuals with a college degree (or higher) among shift workers, whereas almost 1/3 of the day worker had a college degree (or higher). We also observed a significant association between shiftwork status and smoking status. The weighted proportion of current smokers was almost 30% in shift workers, as opposed to 21.56% in day workers.

Finally, our data suggest that body weight is independent of shiftwork status (*p* = 0.717). The weighted proportion of overweight and obese shift workers did not differ significantly from day workers. 

Table 2 shows nutrient and fluid intake of the study population stratified by shiftwork status. We observed no significant intergroup differences with regard to energy intake between both groups (*p* = 0.859). Moreover, mean intake of carbohydrates, protein, and fat did not differ between shift workers and day workers.

Regular day workers consumed significantly more fiber compared to their day working counterparts (16.75 vs. 15.07 g/day). Surprisingly, we observed no differences with regard to caffeine and moisture intake between both groups. Alcohol intake was slightly higher in shift workers; however, the difference was not statistically significant.

Table 3 shows bowel health characteristics of the study population stratified by shiftwork status. Based on our analysis, all examined items (including bowel movement frequency, Bristol Stool Scale, bowel leakage of gas, and fecal incontinence) were independent of shiftwork status. The weighted proportion of individuals suffering from constipation was comparable in both groups (7.09% vs. 6.90%). The weighted proportion of day workers suffering from diarrhea was slightly smaller in the day worker group (6.08% vs. 7.92); however, the differences were not statistically significant. Based on the BSFS assessment, 86.82% of participants in the day worker group and 85.2% in the shift worker group had a normal stool pattern.

Bowel movement frequency was also comparable between both groups. Approximately 62.2% of day workers had 3–7 bowel movements per week. This proportion was smaller in shift workers (59.93%), but the intergroup difference failed to reach statistical significance. Of note, the weighted proportion of individuals with less than three BM (or more than 21 BM) per week in shift workers was very small, and the estimated proportion should be considered unreliable per recent NHCS analytic guidelines.

Table 4 shows the results of the employed multinomial logistic regression model (F(16,1) = 365.49, *p* = 0.0411) investigating potential associations between shiftwork and the BSFS-based stool pattern. Shiftwork status did not significantly affect the odds of having constipation (or diarrhea) relative to a normal stool pattern.

However, the number of clusters limited us in the maximum number of coefficients that could be simultaneously tested. Inclusion of race/ethnicity (which was identified as an important candidate predictor of interest with a bivariate relationship of significance *p* < 0.25) was not possible in this model.

As such, we also ran two simple binary multivariate logistic regression models to ascertain the effects of sex, age, ethnicity, and shiftwork status on the likelihood that participants suffered from constipation (or diarrhea, respectively). Table 5 displays the results for both models (left column: constipation; right column: diarrhea). Both logistic regression models were statistically significant (F(11,6) = 6.21, *p* = 0.018 and F(11,6) = 4.57, *p* = 0.037, respectively). After adjusting for covariates, shift workers did not have significantly higher odds for constipation (or diarrhea) when compared to day workers. Adding income as an additional covariate did not improve the models (F(12,5) = 4.90, *p* = 0.045 and F(12,5) = 3.50, *p* = 0.088), and did not significantly affect the odds for shiftwork status (not shown).

To investigate potential associations between shiftwork status and BM frequency, we employed a cumulative logistic regression model. As mentioned earlier, the weighted proportion of individuals with less than three BM (or more than 21 BM) per week in shift workers was very small, and the estimated proportions were considered unreliable as per recent NHCS analytic guidelines. As such, we recategorized BM frequency for the cumulative logistic regression model. Categories were as follows: less than 7 BM per week, 7–14 BM per week, and more than 14 BM per week. Again, only candidate predictors of interest and with a bivariate relationship of significance (*p* < 0.25) with the response variable were included in this regression model (apart from shiftwork status). Candidate predictors included shiftwork status, age, sex, and race/ethnicity. A significant regression model was found (F(11, 6) = 9.95, *p* = 0.005) and the estimated cumulative odds ratios for the non-reference categories may be obtained from Table 6. The odds of being in one of the higher BM frequency categories for women decreased by about 65% relative to men, whereas no significant difference was found for shift workers (relative to day workers).

## 4. Discussion

We used cross-sectional data from the NHANES (2007–2010) to investigate bowel health in a U.S. population of *n* = 458 shift workers. Our results suggest no association between shiftwork status and all examined bowel health items after adjustment for confounders. Day workers and shift workers did not differ with regard to BM frequency, BSFS, bowel leakage of gas, and FI.

Our results are, to some extent, surprising. Two independent reviews suggested an association between GI disorders and shiftwork [3,8]. In addition to that, several studies dating back to the early 1980s found a higher prevalence of diseases of the digestive tract in shift workers [64]. Segawa et al. reported a higher prevalence of peptic ulcer disease in a Japanese cohort of shift workers [15], and Westerberg and Theorell found shift workers to be overrepresented among dyspepsia patients [65]. Potential associations between shiftwork and GI disorders are summarized in Figure 1.

Of note, not all studies found such associations. Investigations by Dirken et al. and Alfredsson et al. did not demonstrate any significant differences between shift and day workers [66,67].

One factor that may explain the nonsignificant intergroup differences in our sample was nutrient intake in both groups. Apart from fiber intake, nutrient intake did not differ significantly between shift workers and day workers. Although fiber intake differed significantly between both groups (statistically speaking), it is questionable whether this difference was clinically relevant. Fiber intake in both groups was relatively low (16.75 vs. 15.07 g/day), and the lack of a clinically relevant intergroup difference could potentially explain why the groups did not differ with regard to defecation patterns and BM frequency.

In fact, neither of the two groups met the fiber intake recommendations from the Institute of Medicine [68], which range from 19 g to 38 g of fiber per day, depending on sex and age [69]. The amount of daily fiber consumed is an important predictor of stool frequency [70,71,72], and individuals that consume higher amounts of fiber have more bowel movements [73]. The fact that fiber intake in both groups in our sample was comparable (and rather low at the same time) may partially explain the lack of significant intergroup differences.

In addition to that, we observed no significant intergroup differences in other dietary factors that may have influenced bowel movement frequency, such as moisture intake.

Epidemiological evidence from studies in adolescents indicated an association between a lower fluid intake and intestinal constipation [74], although other studies could not confirm this [75]. Moisture intake in our study, however, was similar in both groups. Water supplementation can enhance the effect of a high-fiber diet on stool frequency when sufficient fiber is consumed (at least 25 g/d according to a study by Anti et al. [76]). Notably, this was not the case in our cohort. 

Moreover, we found no differences with regard to caffeine intake between both groups in our sample. Caffeine is a ubiquitous fatigue countermeasure [77] and often used by shift workers to optimize off-duty alertness [78,79]. Caffeine may increase gastrointestinal motility [80] and stimulate a motor response of the distal colon in some people [81], as well as an earlier desire to defecate [82]. Caffeine consumption was also associated with increased odds for inflammatory bowel syndromes in a recent study [83]. The fact that caffeine intake in our cohort did not differ significantly (206.23 vs. 182.20 mg/d, *p* = 0.124) could also explain the lack of associations between shiftwork status and GI symptoms, in particular with regard to BM frequency.

Another dietary factor that warrants consideration is carbohydrate intake. Based on the results of a recently published Chilean study [84], we expected a lower carbohydrate intake in shift workers. A low-carbohydrate intake has been frequently associated with constipation [85,86], and may increase the risk of development of gastrointestinal disorders [85]. Carbohydrate intake in day workers and shift workers in our sample was almost identical (279.14 g/d vs. 277.04 g/d), a fact that could also explain the lack of any significant intergroup differences with regard to bowel health. 

In summary, our data suggested no association between shiftwork status and GI symptoms. Yet, these results warrant caution. To maintain an adequate sample size, we combined evening/night shift workers and rotating shift workers in one group, following the approach of Wirth and colleagues [38]. This approach has some disadvantages, and we clearly acknowledge that different shiftwork schedules may have different effects on human health [87]. Some studies suggested that widely varying work start and end times and evening shifts, in particular, may increase the risk for GI disturbances [88], whereas other studies emphasized the adverse effects of rotating shifts on gastrointestinal health [16]. Our combination does not allow for a more detailed assessment of these relations. Further splitting the groups would have resulted in smaller subgroups with inflated standard errors and unreliable proportion estimates. It is not inconceivable that the investigation of particular subpopulations (e.g., workers that only engage in night shift) would have altered the results of our study. In light of methodological limitations, we refrained from this step.

In addition to that, our study is characterized by some additional limitations. At first, and in light of the available data, we were unable to differentiate soluble and insoluble fiber intake. Insoluble and soluble fibers have different effects on the GI tract [89], and a differentiation would have enhanced the quality of our analysis. Adding fecal biomarkers associated with bowel dysfunction (e.g., fecal bile acids and fecal fat [90]) would have also been an asset to our study, yet those were unavailable in the employed NHANES cycles. More detailed information on the area of activity of our worker sample and their surroundings/work environment would have also been desirable.

Moreover, it is important to emphasize once more that we analyzed cross-sectional data, which do not allow for causal inference. While the sample size is modest, we also acknowledge that some of our proportion estimates were unreliable per NCHS guidelines and must thus be interpreted with caution. Finally, our data date back to the years 2007–2010, which is an additional limitation that has been discussed elsewhere in great detail [28]. Unfortunately, we could not append more recent NHANES cycles (e.g., 2010–2011 and onward) because some essential data on occupational health (for example the question that we used for shiftwork status assessment) were no longer included in these cycles. The United States experienced an economic downturn from 2007 to 2009, and although we adjusted our results for annual household income, this is a fact that has to be kept in mind when interpreting our results in a larger context.

On the other hand, our study also draws upon a number of strengths. Our data stem from a nationally representative dataset (National Health and Nutrition Examination Survey) that has been often used to investigate health-related questions in shift-working populations [23,24,25,26,27,28]. Research investigating GI health in shift workers using U.S. data is generally scarce, and we tried to address this gap in the literature with our study. As mentioned in our previous study [49], the employed assessment methods were found to be valid and reliable, which is important when comparing different populations.

Additional studies may address the weaknesses of this cross-sectional investigation and could also investigate food groups and biomarkers that may help to gain additional insights into GI health in shift workers. As mentioned earlier, additional studies should also focus on the different shiftwork patterns (e.g., night shifts only vs. rotation shifts) and consider potential differences in their health impacts.

## 5. Conclusions

The present study sought to explore bowel health in a United States-based population of shift workers. Despite some significant (but clinically irrelevant) intergroup differences with regard to nutrient intake (e.g., daily fiber consumption), GI health did not differ between both groups. We observed no difference with regard to several key nutrients (e.g., carbohydrate intake, caffeine intake, etc.) that influence BM frequency and defecation patterns. The low fiber intake in both groups could be a key factor explaining our findings, yet causal interference is impossible due to the cross-sectional nature of our data. Additional studies in this field are warranted, preferably including food group analyses and fecal biomarkers.

## Figures and Tables

**Figure 1 ijerph-19-03334-f001:**
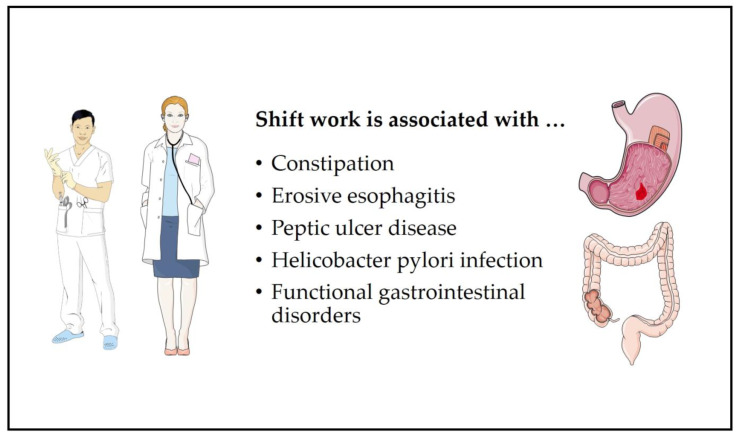
Shiftwork and its associations with gastrointestinal disorders, based on [12,13,14,15,16]. Modified from Servier Medical Art database by Servier (www.smart.servier.com (accessed on 8 March 2022); Creative Commons 3.0).

**Figure 2 ijerph-19-03334-f002:**
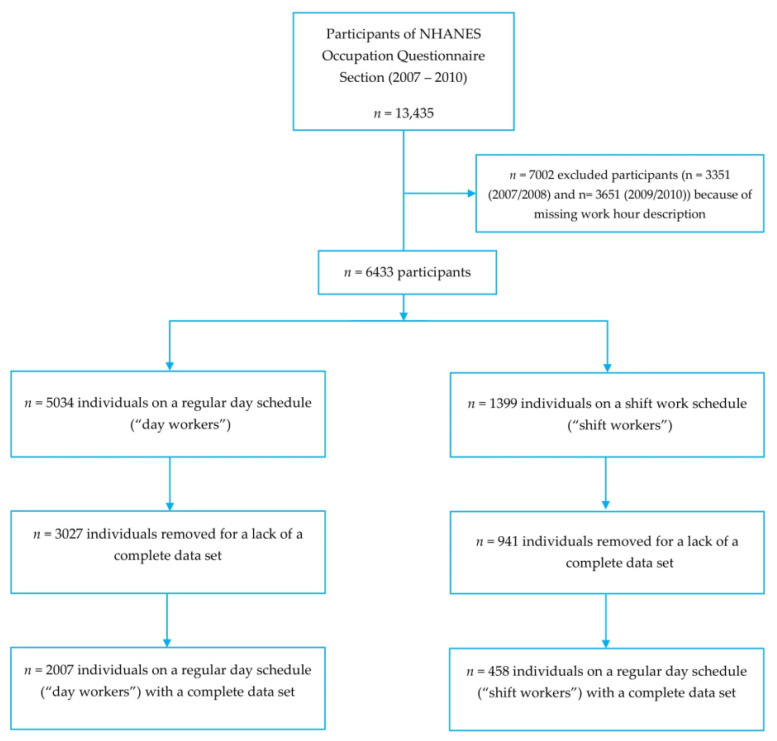
Inclusion flow chart: NHANES 2007–2010.

**Table 1 ijerph-19-03334-t001:** Study sample characteristics: a comparison by shiftwork status.

	Day Workers (*n* = 2007)	Shift Workers (*n* = 458)	*p*-Value
Sex			
Male	*n* = 1081 (53.69%(1.41))	*n* = 252 (54.36%(3.27))	0.857
Female	*n* = 926 (46.31%(1.41))	*n* = 206 (45.64%(3.27))	
Age			
Mean (SE)	42.53 (0.45)	37.89 (0.61)	<0.001
Race/Ethnicity			
Mexican American	*n* = 418 (9.44%(1.43))	*n* = 88 (9.41%(1.59))	0.003
Other Hispanic	*n* = 239 (4.84%(1.04))	*n* = 54 (5.84%(2.14)) ^c^	
Non-Hispanic White	*n* = 904 (70.33%(3.43))	*n* = 172 (63.77%(4.42)) ^a^	
Non-Hispanic Black	*n* = 364 (9.32%(1.78))	*n* = 129 (16.45%(2.79)) ^a^	
Other race	*n* = 81 (6.08%(0.96))	*n* = 15 (4.53%(1.25))	
Marital status			
Married/living with partner	*n* = 1355 (70.65%(2.10))	*n* = 233 (50.93%(3.12)) ^a^	<0.001
Widowed/divorced/separated	*n* = 320 (13.54%(1.03))	*n* = 83 (15.52%(2.35))	
Never married	*n* = 332 (15.81%(1.62))	*n* = 142 (33.55%(1.83)) ^a^	
Education level			
Less than 9th grade	*n* = 149 (3.47%(0.61))	*n* = 47 (5.20%(0.72))	<0.001
9–11th grade	*n* = 308 (10.97%(1.30))	*n* = 84 (14.56%(2.18))	
High School Grad/GED	*n* = 454 (22.10%(1.54))	*n* = 128 (30.30%(2.60)) ^a^	
Some college or AA degree	*n* = 581 (30.43%(1.32))	*n* = 145 (35.83%(2.50))	
College graduate or above	*n* = 515 (33.03%(2.60))	*n* = 54 (14.11%(1.38)) ^a^	
Annual household income			
Under USD 20,000	*n* = 221 (6.64%(0.76))	*n* = 92 (15.37%(1.25)) ^a^	<0.001
Over USD 20,000	*n* = 1786 (93.36%(0.76))	*n* = 366 (84.63%(1.25)) ^a^	
Smoking status			
Never smoker	*n* = 1140 (56.63%(1.44))	*n* = 246 (52.56%(3.40))	0.022
Former smoker	*n* = 425 (21.81%(1.10))	*n* = 85 (17.74%(2.73))	
Current smoker	*n* = 442 (21.56%(1.36))	*n* = 127 (29.70%(2.63)) ^a^	
Body weight			
Underweight	*n* = 20 (0.92%(0.29))	*n* = 6 (1.37%(0.63)) ^c^	0.717
Normal weight	*n* = 564 (30.49%(1.06))	*n* = 131 (30.46%(3.32))	
Overweight	*n* = 716 (36.01%(1.63))	*n* = 148 (32.84%(4.21))	
Obesity	*n* = 707 (32.58%(1.73))	*n* = 173 (35.32%(4.07))	

Table 1 legend: Column percentages may not equal 100% due to rounding. The *p*-value is based on STATA’s design-based Rao–Scott *F*-test and tests for a potential association between shiftwork status and the respective variable (categorical variables only). ^a^: Indicates significant differences in the weighted proportions; ^c^: estimate considered unreliable per NHCS analytic guidelines and based on STATA’s postestimation command “kg_nchs”.

**Table 2 ijerph-19-03334-t002:** Nutrient and fluid intake: a comparison by shiftwork status.

	Day Workers (*n* = 2007)	Shift Workers (*n* = 458)	*p*-Value
Calories (kcal)/day	2360.01 (37.84)	2342.87 (71.80)	0.859
Protein (g/day)	90.14 (1.79)	87.88 (3.36)	0.598
Carbohydrate (g/day)	279.14 (3.37)	277.04 (9.42)	0.842
Fat (g/day)	91.31 (2.03)	89.66 (3.26)	0.717
Saturated fat (g/day)	30.02 (0.72)	30.46 (1.50)	0.824
Monounsaturated fat (g/day)	33.91 (0.82)	33.00 (1.22)	0.598
Polyunsaturated fat (g/day)	19.54 (0.45)	18.47 (0.59)	0.188
Fiber (g/day)	16.75 (0.58)	15.07 (0.45)	0.009
Alcohol (g/d)	12.81 (1.03)	14.19 (1.82)	0.538
Caffeine (mg/d)	206.23 (8.92)	182.20 (18.90)	0.124
Moisture (g/d)	3182.84 (64.40)	3214.43 (101.47)	0.803

Table 2 legend: For continuous and normally-distributed variables, data are shown as mean (and standard error in parenthesis). Continuous variables were compared using STATA’s lincom (linear combinations of estimator) post-estimation command.

**Table 3 ijerph-19-03334-t003:** Bowel health characteristics: a comparison by shiftwork status.

	Day Workers (*n* = 2007)	Shift Workers (*n* = 458)	*p*-Value
Bristol Stool Scale			
BSFS Type 1	*n* = 45 (2.19%(0.40))	*n* = 6 (1.31%(0.79)) ^c^	0.335
BSFS Type 2	*n* = 95 (4.90%(0.73))	*n* = 30 (5.58%(0.82))	
BSFS Type 3	*n* = 540 (27.72%(0.97))	*n* = 121 (29.08%(2.39))	
BSFS Type 4	*n* = 1041 (51.56%(1.76))	*n* = 220 (47.33%(2.54))	
BSFS Type 5	*n* = 163 (7.54%(1.05))	*n* = 40 (8.76%(1.41))	
BSFS Type 6	*n* = 117 (5.77%(0.68))	*n* = 36 (6.79%(1.36))	
BSFS Type 7	*n* = 6 (0.32%(0.15)) ^c^	*n* = 5 (1.13%(0.64)) ^c^	
Stool pattern (BSFS-based)			
Constipation	*n* = 140 (7.09%(0.79))	*n* = 36 (6.90%(1.21))	0.396
Normal	*n* = 1744 (86.82%(1.06))	*n* = 381 (85.18%(2.22))	
Diarrhea	*n* = 123 (6.08%(0.67))	*n* = 41 (7.92%(1.56))	
Bowel Movement Frequency			
<3/week	*n* = 51 (2.50%(0.37))	*n* = 19 (4.91%(1.82)) ^c^	0.252
3–7/week	*n* = 1227 (64.20%(1.07))	*n* = 267 (59.93%(2.40))	
8–14/week	*n* = 590 (26.53%(1.15))	*n* = 141 (28.28%(2.02))	
≥15–21/week	*n* = 117 (5.68%(0.68))	*n* = 25 (5.80%(1.33))	
≥21/week	*n* = 22 (1.08%(0.27))	*n* = 6 (1.08%(0.71)) ^c^	
Bowel leakage: gas			
2 or more times a day	*n* = 214 (10.57%(1.26))	*n* = 37 (9.42%(1.65))	0.768
Once a day	*n* = 182 (9.50%(0.82))	*n* = 30 (8.52%(1.36))	
2 or more times a week	*n* = 132 (6.98%(0.70))	*n* = 22 (5.66%(1.02))	
Once a week	*n* = 104 (5.79%(0.36))	*n* = 21 (5.97%(2.23)) ^c^	
1–3 times a month	*n* = 212 (10.87%(0.65))	*n* = 49 (10.63%(1.62))	
Never	*n* = 1163 (56.29%(1.13))	*n* = 299 (59.80%(2.75))	
Fecal incontinence (FISI)			
Yes	*n* = 25 (1.16%(0.37)) ^c^	*n* = 26 (1.36%(0.70)) ^c^	0.817
No	*n* = 1982 (98.84%(0.70))	*n* = 452 (98.64%(0.70)))	

Table 3 legend: Column percentages may not equal 100% due to rounding. The *p*-value is based on STATA’s design-based Rao–Scott *F*-test and tests for a potential association between shiftwork status and the respective variable. ^c^ = Estimate considered unreliable per NHCS analytic guidelines (based on STATA’s postestimation command “kg_nchs”).

**Table 4 ijerph-19-03334-t004:** Multinomial logistic regression: Estimates of adjusted odd ratios for the stool pattern outcome.

	Constipation: Normal	*p*	Diarrhea: Normal	*p*

Sex				
Female	2.72 (1.59–4.68)	0.001	1.26 (0.78–2.04)	0.314
Male	-		-	
Age				
18–24 years	-		-	
25–34 years	0.32 (0.18–0.62)	0.002	2.08 (1.05–4.15)	0.038
35–44 years	0.49 (0.29–0.82)	0.009	2.49 (1.40–4.43)	0.004
45–54 years	0.44 (0.21–0.89)	0.026	2.23 (1.18–4.23)	0.017
55–64 years	0.31 (0.17–0.55)	<0.001	2.77 (1.36–5.65)	0.008
>65 years	0.24 (0.09–0.64)	0.007	3.82 (2.17–6.73)	0.000
Annual household income				
Under USD 20,000	-		-	
Over USD 20,000	1.12 (0.59–2.12)	0.723	0.62 (0.40–0.95)	0.032
Shiftwork status				
Day workers	-		-	
Shift workers	0.85 (0.55–1.33)	0.455	1.40 (0.88–2.22)	0.142

Table 4 legend: OR are displayed with their 95% confidence intervals and *p*-value. The symbol “-” indicates the reference category. *p* = *p*-value.

**Table 5 ijerph-19-03334-t005:** Logistic regression models investigating potential associations between shiftwork status and stool patterns (constipation and diarrhea).

	Constipation	*p*	Diarrhea	*p*
Sex				
Female	2.61 (1.51–4.52)	0.002	1.20 (0.75–1.94)	0.416
Male	-		-	
Age				
18–24 years	-		-	
25–34 years	0.32 (0.17–0.60)	0.002	2.24 (1.18–4.29)	0.017
35–44 years	0.47 (0.28–0.79)	0.007	2.59 (1.47–4.55)	0.003
45–54 years	0.43 (0.20–0.91)	0.031	2.34 (1.28–4.30)	0.009
55–64 years	0.29 (0.17–0.52)	<0.001	3.01 (1.48–6.13)	0.005
>65 years	0.21 (0.08–0.59)	0.005	4.24 (2.31–7.81)	<0.001
Ethnicity				
Mexican American	0.78 (0.52–1.17)	0.218	1.20 (0.78–1.85)	0.370
Other Hispanic	1.15 (0.69–1.91)	0.561	1.53 (1.04–2.24)	0.032
Non-Hispanic White	-	-	-	-
Non-Hispanic Black	1.57 (0.91–2.68)	0.097	0.61 (0.33–1.15)	0.120
Other race	0.45 (0.20–1.01)	0.053	0.81 (0.28–2.36)	0.693
Shiftwork status				
Day workers	-		-	
Shift workers	0.80 (0.53–1.20)	0.260	1.51 (0.97–2.37)	0.067

Table 5 legend: OR are displayed with their 95% confidence intervals and *p*-value. The symbol “-” indicates the reference category. *p* = *p*-value.

**Table 6 ijerph-19-03334-t006:** Estimated cumulative odds ratios in the cumulative logistic regression model for bowel movement frequency.

	Cumulative Odds Ratio	*p*
Sex		
Female	0.35 (0.24–0.52)	<0.001
Male	-	-
Age		
18–24 years	-	-
25–34 years	0.95 (0.59–1.52)	0.809
35–44 years	1.58 (1.03–2.42)	0.039
45–54 years	1.66 (0.98–2.80)	0.058
55–64 years	1.41 (0.86–2.30)	0.151
>65 years	2.37 (1.45–3.89)	0.002
Mexican American	1.59 (1.14–2.21)	0.010
Other Hispanic	1.01 (0.55–1.83)	0.995
Non-Hispanic White	-	-
Non-Hispanic Black	0.81 (0.58–1.12)	0.182
Other race	0.98 (0.51–1.87)	0.947
Shiftwork status		
Day workers	-	-
Shift workers	1.07 (0.73–1.57)	0.704

Table 6 legend: OR are displayed with their 95% confidence intervals and *p*-value. The symbol “-” indicates the reference category. *p* = *p*-value.

## Data Availability

Data are publicly available online (https://wwwn.cdc.gov/nchs/nhanes/Default.aspx (accessed on 11 February 2022). The datasets used and analyzed during the current study are available from the corresponding author on reasonable request.

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
