# Peer review of "Bowel Health in U.S. Shift Workers: Insights from a Cross-Sectional Study (NHANES)"

_ijerph, 2022, doi:10.3390/ijerph19063334_

Round 1

Reviewer 1 Report

The paper is about the comparison of two groups of US shift workers (regular daytime and outside that regular schedule) and their differences in terms of bowel health and related aspects or issues.

It is an interesting paper on a neglected aspect related to working conditions and health.

Is there any relation that can be made to space (urban space and rural areas)?

The paper could specify more the core of the research and justify the choice made for the research design. It would be helpful to justify why the authors study that data in particular? 

Is there any other cohort or cycle than 2007-2008 and 2009-2010? Are the surveys compatible in terms of questions asked? Was there any change in wording or nomenklature? Why not other more recent data? Problem of correspondence? This is a limitation of the study,

I would have expected more discussion of table 1.

What is the difference between the two cycles or cohorts used? There is not much presentation of the features of each cohort and each group for each cohort or cycle. What was the sampling technique used in each cycle and context of the survey and information similarity between the two studies. 

What external factors could have been discussed. What confounding factors could be discussed or considered in further research?

I consider that the discussion could be a little better and the conclusion very short. 

Could you clarify the links between this paper and another paper published here Nutrients | Free Full-Text | Bowel Health in U.S. Vegetarians: A 4-Year Data Report from the National Health and Nutrition Examination Survey (NHANES) | HTML (mdpi.com)

https://www.mdpi.com/2072-6643/14/3/681/htm 

Author Response

Dear Reviewer,

We would like to thank you very much for careful and thorough reading of this manuscript and for the thoughtful comments and constructive suggestions, which help to improve the quality of this article. We made all the requested revisions to our original manuscript based on all the comments we received from you. All changes have been clearly marked in yellow and blue color. We appreciate your valuable input and time. Sincerely,

The authors

Reviewer 2 Report

Dear Authors; This paper covers a very interesting topic in public health. However, its current version needs some serious work. Regards.

P.S.

[1] Statistical

1-1 The Statistical Analysis Plan(SAP) presented in this manuscript results section in insufficient and in the statistical community is a mere "Descriptive Analysis".  To investigate properly the associations of interest, one needs to fit  logistic regression models, multinomial regression models, and Poisson regression models with outcomes of interest in the left column of Table 3 and predictor "Shift Status" controlled for other covariates. 

Link: https://en.wikipedia.org/wiki/Multinomial_logistic_regression

Link: https://en.wikipedia.org/wiki/Poisson_regression

So, the current results should be divided into two subsections: 4.1. Descriptive Results; 4.2. Regression Model results.  Whatever text is in the current manuscript version results section goes into subsection 4.1. Then, the regression model results including the model and plots of interest  go into subsection 4.2. 

Also, the math formulaes for the regression models need to be added in the Statistical Analysis section 2.3 and the entire abstract and Discussion section of the paper need to be rewritten. 

1-2 The data is relatively outdated for 14 years ago and in the time of USA economic recession.  The economic status is itself a confounder here. I am skeptic how much this distorts the results. It was better to have data for more "normal times" say in 2015-2016 or so. Perhaps it is good the authors mention this issue in the Discussion section limitations.

1-3 Lin 162 STAT software needs its additional reference in the References section.

[2] Writing

2-1 It is recommended the authors add list of used abbreviations right before reference section for the readers referral. This journal is international journals and readers deserve to have minimum effort to get access to the items frequently used in the paper. Example.

Abbreviations

BM   Bowel Movement

GI   gastrointestinal 

Author Response

(The authors gave the same response as above.)

Round 2

Reviewer 2 Report

Dear Authors; most of my concerns were addressed satisfactorily. Regards.